# Preoperative Health-Related Quality of Life Predicts Minimal Clinically Important Difference and Survival after Surgical Resection of Hepatocellular Carcinoma

**DOI:** 10.3390/jcm8050576

**Published:** 2019-04-27

**Authors:** Chong-Chi Chiu, King-Teh Lee, Jhi-Joung Wang, Ding-Ping Sun, Hao-Hsien Lee, Chien-Cheng Huang, Hon-Yi Shi

**Affiliations:** 1Department of General Surgery, Chi Mei Medical Center, Liouying 73657, Taiwan; chiuchongchi@yahoo.com.tw (C.-C.C.); hao_hsien@hotmail.com (H.-H.L.); 2Department of General Surgery, Chi Mei Medical Center, Tainan 71004, Taiwan; sdp0127@gmail.com; 3Department of Electrical Engineering, Southern Taiwan University of Science and Technology, Tainan 71005, Taiwan; 4Division of Hepatobiliary Surgery, Department of Surgery, Kaohsiung Medical University Hospital, Kaohsiung 80708, Taiwan; ktlee@kmu.edu.tw; 5Department of Healthcare Administration and Medical Informatics, Kaohsiung Medical University, Kaohsiung 80708, Taiwan; 6Department of Medical Research, Chi Mei Medical Center, Tainan 71004, Taiwan; 400002@mail.chimei.org.tw; 7Department of Emergency Medicine, Chi Mei Medical Center, Tainan 71004, Taiwan; honyi6165@yahoo.com; 8Department of Senior Services, Southern Taiwan University of Science and Technology, Tainan 71005, Taiwan; 9Department of Business Management, National Sun Yat-sen University, Kaohsiung 80424, Taiwan; 10Department of Medical Research, Kaohsiung Medical University Hospital, Kaohsiung 80708, Taiwan

**Keywords:** hepatocellular carcinoma, health-related quality of life, minimal clinically important difference, survival

## Abstract

Despite the growing use of minimal clinically important difference (MCID) as a cancer outcome measure, no study has reported clinically significant outcomes in cancer patients. We defined MCID and evaluated the use of preoperative HRQoL for predicting MCID and survival after surgical resection of hepatocellular carcinoma (HCC). In total, 369 patients completed the Functional Assessment of Cancer Therapy-Hepatobiliary (FACT-Hep) and the SF-36 at baseline and at two years post-operative at three tertiary academic hospitals. The corresponding MCID values were 3.6 (SF-36 physical component summary), 4.2 (SF-36 mental component summary), 5.4 (FACT-General total score), and 6.7 (FACT-Hep total score). The predictors of achieving postoperative MCID were significantly higher in patients who had low preoperative HRQoL score, advanced age, high education level, and high BMI (*p* < 0.05). However, patients with a high preoperative HRQoL score, high education level, high BMI, and low Charlson comorbidity index score were significantly associated with survival (*p* < 0.05). Preoperative HRQoL scores were predictive of MCID and overall survival after surgical resection of HCC. The findings of this study may be useful for managing the preoperative expectations of candidates for HCC resection and for developing shared decision-making procedures for patients undergoing surgical resection of HCC.

## 1. Introduction

Hepatic resection is the mainstay curative treatment for patients with hepatocellular carcinoma (HCC), even in some patients with early-stage HCC [1,2,3]. Health-related quality of life (HRQoL) is a recognized indicator of healthcare outcomes and, since the 1990s, evaluations of cancer treatment outcomes have increasingly emphasized assessment of HRQoL [4,5]. Disease-specific and generic HRQoL measures are often reported together and provide complementary assessments of patient well-being before and after an intervention. It is important not to mix up the concept of quality of life with a recently growing area of HRQoL. Quality of life is an essential concept in the field of international development since it allows analysis of development on a measure broader than the standard of living [6,7]. Within development theory, however, there are varying ideas concerning what constitutes desirable change for a particular society, and the different ways that institutions define the quality of life, therefore, shapes how these organizations work for its improvement as a whole. The Functional Assessment of Cancer Therapy-Hepatobiliary (FACT-Hep) measure is one of the most widely used patient-reported questionnaires for measuring HRQoL in cancer research [8]. Unfortunately, research has shown that most of the studies that have used the FACT-Hep lack any reporting of clinical significance, even though guidelines for assessing clinical significance do exist.

A clinically important difference is a change that a patient or clinician would consider meaningful or worthwhile, such that an intervention or treatment would be considered worthy of repeating or such that patients would consider the change as an improvement in HRQoL. As such, measures of clinical significance such as the minimal clinically important difference (MCID) are increasingly used as a standard clinical outcome measure [6,7]. The MCID is defined as the smallest outcome change that the patient perceives as clinically important [9]. Despite the growing use of MCID as a cancer outcome measure, no study has reported clinically significant outcomes after surgical resection of HCC.

For cancer patients and their families, clinical data for HRQoL outcomes provide a useful indicator of the expected course of recovery and the expected effects of treatment. Thus, HRQoL data can help them make informed treatment decisions [10,11]. Baseline assessments of HRQoL have proven useful for predicting survival in various cancers, including colorectal, esophageal, breast, oropharyngeal, and lung cancers [10,11,12,13]. For varying severity of cancer, HRQoL has shown higher sensitivity compared to conventional prognostic indicators and compared to physician assessments [11,14]. Gotay et al. assessed the use of patient-reported HRQoL as a prognostic indicator of cancer outcomes [15]. Out of 39 clinical trials reviewed by the authors, 36 reported at least one HRQoL domain that was a significant predictor of survival. However, comparisons of results published in the literature are difficult because studies differ in the HRQoL measures applied and studied populations differ in patient attributes such as the type, site, and stage of disease [14,15,16,17]. Given the variability in reported overall survival rates and overall study heterogeneity, the evidence base for overall survival and determinants of overall survival after surgical resection of HCC are still evolving, and continued investigation is warranted.

A growing body of evidence indicates that preoperative functional status or HRQoL are important determinants of cancer surgery outcomes [14,15,16,17]. To the best of our knowledge, no prior study has systematically evaluated the role of preoperative HRQoL in achieving MCID and in overall survival after surgical resection of HCC. Therefore, the purpose of this study was to investigate the use of preoperative HRQoL scores for predicting achievement of MCID and for predicting overall survival after surgical resection of HCC.

## 2. Materials and Methods

### 2.1. Subjects and Data Collection

This study recruited all patients who had received surgical resection of HCC performed at one of three southern Taiwan medical centers between February 2013 and February 2017. For accurate assessment of postoperative outcome measures, the analysis was limited to patients who had received surgical resection performed by a director of surgery in a medical institution or by a senior attending doctor specializing in HCC surgery or treatment. Inclusion criteria were the following: (1) a histologic or combined radiographic and laboratory diagnosis of HCC, (2) ability to communicate in Chinese or Taiwanese, and (3) agreement to participate in a questionnaire survey performed in the hospital ward or by telephone. Major exclusion criteria included concurrent malignancy or participation in another quality-of-life study that might have interfered with this study. Figure 1 shows that, during the sample selection period, 496 subjects were eligible for participation. Of these, 62 were excluded due to benign tumor or cognitive impairment. Therefore, 369 subjects were assessed preoperatively (baseline) and at 2 years postoperatively. Baseline demographic and clinical data were collected through questionnaire surveys and medical records reviews. This study was approved by the Institutional Review Board of Chi Mei Medical Center (10002-L01).

### 2.2. Study Protocol

Patients were asked to complete the questionnaires during follow-up visits at our outpatient clinic. To maximize compliance and minimize volunteer bias, a research assistant was available to help patients complete the questionnaires during each outpatient session. All HRQoL data were collected by the same research assistant. Patients were informed that their questionnaire responses would not be revealed to their attending surgeons and, hence, would not affect their treatment.

### 2.3. Measures of HRQoL

The Short Form-36 (SF-36) Health Survey measures eight dimensions: physical function, role limitation due to physical health, bodily pain, general health, vitality, social function, role limitation due to emotional health, and mental health. To compare the overall physical and mental functioning of the study population with those in the general Taiwan population, physical component summary scores (PCS) and mental component summary scores (MCS) were calculated by norm-based scoring methods and used as dependent variables [18]. Based on a previous study [19], the PCS and MCS were computed in comparison with the general population of Taiwan. Values below 50 indicated that the examined PCS or MCS were below the average values for the general Taiwan population, and vice versa.

The FACT-Hep measure contains five dimensions: physical well-being, social/family well-being, functional well-being, emotional well-being, and additional concerns. The subscales for the physical well-being, social/family well-being, and functional well-being dimensions each contained seven items with a subscale score range of 0–28 points; the subscale for emotional well-being contained six items with a subscale score range of 0–24 points; the subscale for additional concerns about HCC contained 18 items with a subscale score range of 0–72 points [8]. The Functional Assessment of Cancer Therapy-General (Fact-G) and additional concerns for HCC scores were summed to obtain the FACT-Hep total score, which ranged from 0 to 180. Higher scores on all FACT-Hep dimensions were interpreted as better HRQoL and fewer symptoms.

### 2.4. Statistical Analysis

Studies show that a distribution-based method can reliably derive MCID calculated as one-half the standard deviation (SD) in outcome score change from baseline to the two-year follow up for a given instrument in a patient cohort [9,20]. Therefore, this methodology was used to determine MCID values for the SF-36 PCS, SF-36 MCS, FACT-G total score, and FACT-Hep total score.

Multivariable logistic regression models were used to identify predictors of the achievement of MCID after surgical resection of HCC. A Cox multivariable proportional hazard regression model was also used to evaluate how other prognostic factors affect survival. Survival distribution was estimated by Kaplan–Meier method. Significant differences in survival probability were stratified by a log-rank test. Hazard ratios (HRs) and 95% confidence intervals (CIs) were estimated from regression coefficients.

Variables included in the multivariable analyses were gender, age, marital status, education, body mass index (BMI), Charlson co-morbidity index (CCI) score, co-residence with family, smoking, drinking, tumor stage, chemotherapy, radiotherapy, and average length of stay (ALOS). Multivariable analyses also included preoperative HRQoL score. Variables that fell out of the model were excluded from the tables of multivariable results. Statistical analyses were performed using SPSS software (IBM SPSS Statistics for Windows, Version 20.0, Armonk, NY, USA). All statistical tests were two tailed with a significance level of 0.05.

## 3. Results

### 3.1. Patient Demographics

The SF-36 and FACT-Hep measures were completed by 369 HCC surgery patients preoperatively and at two years postoperatively. We compared the patients who remained in the study throughout the two-year period with those who were lost or dead to follow up between the baseline and the second year after discharge. There was no difference in terms of gender, age, marital status, education, BMI, CCI score, co-residence with family, smoking, drinking, tumor stage, chemotherapy, radiotherapy, ALOS, or in any preoperative HRQoL parameters mentioned above (data not shown). Table 1 presents their demographic and clinical characteristics.

### 3.2. HRQoL Outcomes

The patients had a mean age of 60.2 ± 10.8 years, and 73.4% (271) patients were male. Table 2 shows that mean patient-reported HRQoL scores at two years after surgery were significantly higher than those before surgery (*p* < 0.001). The MCID values were 3.6 for the SF-36 PCS; 4.2 for the SF-36 MCS; 5.4 for the FACT-G total score; and 6.7 for the FACT-Hep total score.

### 3.3. Multivariable Analyses

Multivariable analyses of HRQoL and survival were performed to identify predictors of the achievement of MCID after surgical resection for HCC. For each HRQoL measure, a high preoperative score negatively predicted achievement of MCID (*p* < 0.001) (Table 3). According to the SF-36 PCS data, the odds of achieving MCID were lower in males than in females (odds ratio (OR), 0.31; 95% CI, 0.12–0.83) but were higher in patients with advanced age (OR, 1.05; 95% CI, 1.01, 1.10), high education level (OR, 1.14; 95% CI, 1.01, 1.30), and high BMI (OR, 1.11; 95% CI, 1.09, 1.12) compared to their counterparts with young age, low education level, and low BMI, respectively. According to the SF-12 MCS data, the odds of achieving MCID were lower in patients with high BMI compared to those with low BMI (OR, 0.91; 95% CI, 0.84, 0.99); however, the odds of achieving MCID were higher in those with advanced age (OR, 1.01; 95% CI, 1.01, 1.02) and high CCI score (OR, 1.53; 95% CI, 1.13, 1.94) compared to their counterparts with young age and low CCI score, respectively. According to the FACT-G total data, the odds of achieving MCID were higher in patients with advanced age (OR, 1.04; 95% CI, 1.01, 1.07), high education level (OR, 1.12; 95% CI, 1.04, 1.21), and high BMI (OR, 1.19; 95% CI, 1.09, 1.29) compared to their counterparts with young age, low education level, and low BMI, respectively. According to the FACT-Hep total data, the odds of achieving MCID were higher in patients with advanced age (OR, 1.05; 95% CI, 1.02, 1.07), high education level (OR, 1.11; 95% CI, 1.03, 1.19), and high BMI (OR, 1.09; 95% CI, 1.01, 1.18) compared to their counterparts with young age, low education level, and low BMI.

Multivariate analyses of each pre-operative HRQoL score showed that a high score was a positive predictor of overall survival (Table 4). Lower pre-operative HRQoL scores were significantly associated with post-operative morbidity (*p* < 0.05). Education level, BMI, and CCI also showed significant associations with overall survival (*p* < 0.001).

## 4. Discussion

This study investigated how patient-reported preoperative HRQoL affects two outcomes of surgical resection of HCC: MCID and overall survival. Multivariate analyses showed that each preoperative HRQoL score was predictive of both MCID and overall survival. Low preoperative HRQoL score, advanced age, high education level, and high BMI were significantly associated with achievement of postoperative MCID (*p* < 0.05). Additionally, high preoperative HRQoL score, high education level, high BMI, and low CCI had significant positive associations overall survival (*p* < 0.05). It demonstrates that, at baseline, preoperative HRQoL scores relates to postoperative mortality. Lower scores in physical and functional domains are associated with an increased risk of postoperative mortality. The importance of preoperative HRQoL scores for predicting outcomes of surgical resection in HCC patients is now well recognized [9,21]. The current study found that, for a given HRQoL outcome measure, a high preoperative score was significantly for not achieving a postoperative MCID in the outcome measure. The likely explanation for this finding is that patients who already have high HRQoL scores before surgery and less potential for achieving a HRQoL score improvement that meets the criteria for an MCID.

This study aimed to calculate and report the MCID value of commonly used HRQoL scales. Changes in HRQoL by time and/or treatment may not correlate with the direction (positive vs. negative) as well as the magnitude of clinical improvements in outcomes as they are perceived by the patients. Furthermore, cancer treatment has a more significant impact on HRQoL among HCC surgical patients. The MCID for (SF-36 PCS, SF-36 MCS, and FACT-G total score) value changes differed across domains, and they differed for perceived improvement and deterioration. We knew that domain scores related to physical function diminished from pre-treatment to on- or immediately after treatment, and emotional function improved. Additionally, the initial anxiety of the diagnosis and treatment initiation period may have been ameliorated by subsequent familiarity and supportive psychosocial care provided by the clinical service teams after patients finished the initial questionnaire. Thus, this might explain why those patients with a low preoperative score did achieve a postoperative MCID in the outcome measure and regarded as “efficacy” when compared with that achieved by those patients who had full pre-operative familiarity and mental support and thus noted with a high preoperative score.

The significant associations revealed by the HRQoL instruments investigated in this study underscore the relationship between HRQoL measures and medical outcomes. The study showed that preoperative HRQoL scores accurately predict postoperative MCID and overall survival, which is consistent with the literature [11,15,17]. Therefore, counseling is essential for apprising HCC resection candidates of expected postoperative improvements and impairments. If medical outcomes are considered benchmarks, then the preoperative HRQoL score, which is an important predictor of postoperative MCID and overall survival, is crucial.

Until now, no studies have described the implications of significant changes in disease-specific and generic HRQoL outcome measures in patients who have undergone HCC surgery. Steel et al. evaluated the clinical meaningfulness of FACT-Hep scores in HCC patients [22]. The authors combined distribution-based analyses with cross-sectional anchor-based analyses to obtain minimally important differences (MIDs) in FACT-G subscale scores (2–3 points), FACT-G total scores (6–7 points), Hepatobiliary Cancer Subscale scores (MID 5–6 points), and FACT-Hep scores (MID 8–9 points). However, data for clinically significant improvements in HCC surgery outcomes, particularly patient-reported outcomes, are very limited [4,5,6]. For patients who undergo surgical resection of HCC, the current study obtained MCID values of 3.6 for SF-36 PCS; 4.2 for SF-36 MCS; 5.4 for FACT-G total score; and 6.7 for FACT-Hep total score. These data are a novel addition to the literature and provide a useful reference for further studies of HRQoL outcomes after surgical resection of HCC. Notably, this study used patient-reported data for the period from before surgery until two years after surgery. A distribution-based method was used to calculate MCID. Since MCID values may change depending on the time point studied and the psychometric method used for analysis, further studies are needed to investigate MCID after surgical resection of HCC using different time points and mixed anchor/distribution-based methods of deriving MCID values.

The evidence base for overall survival after surgical resection of HCC is growing but is still relatively limited. The overall survival rates reported in the literature are somewhat variable, and the studied populations have been heterogeneous. Quinten et al. investigated the prognostic relationship between HRQoL and survival in a dataset for 30 randomized controlled trials performed by the European Organization for Research and Treatment of Cancer [11]. Their study showed that, in each cancer site, at least one HRQoL domain had an additive prognostic value that exceeded the prognostic values of clinical and sociodemographic variables. A systematic literature review by Quinten et al. confirmed that baseline HRQoL and at least one HRQoL domain were significantly associated with overall survival [11]. The current study found that preoperative HRQoL scores, education level, and BMI had significant positive associations with overall survival (*p* < 0.05) whereas CCI score had a significant negative association with overall survival (*p* < 0.05).

The HRQoL factors identified in this study varied from those in previous studies [9,20,22]. One possible explanation is differences in study populations. Our study focused on patients in both early and advanced stages of HCC whereas previous studies have only focused on patients in advanced stages of the disease. Secondly, patients with different cultural backgrounds may have different perceptions of HRQoL. Thirdly, patients in recent studies have more treatment options compared to patients in earlier studies, which can result in different perceptions of the implications of HCC and thus different perceptions of HRQoL. Fourthly, even studies that use the same HRQoL measure may have very different data analysis methodologies.

The findings of this study have important implications for preoperative counseling of patients, management of patient expectations, and stratification of outcomes. Healthcare providers increasingly emphasize shared decision making and are now using predictive modeling to inform surgery patients about potential clinical outcomes and the likelihood of success [11,22]. The results of the present study suggest that healthcare providers should consider routinely administering HRQoL instruments preoperatively as screening tools and for informing shared decision-making strategies. Patients with low HRQoL scores can be referred for counseling to modify their outcome expectations or referred for targeted interventions to optimize their HCC resection outcomes.

Certain limitations of this study are noted. Firstly, the patient data were derived from a multi-institutional HCC registry containing data contributed by multiple surgeons. As such, other than institutional best practices, surgical techniques and rehabilitation protocols were not standardized. However, since patient-reported HRQoL outcomes were obtained at the time of each clinical encounter, the reports of functional status are assumedly accurate. Another limitation is that sensitivity of the MCID values was not analyzed. The MCID can be calculated according to a consensus of or by using an anchor-based method or a distribution-based method [9,23]. Each methodology for deriving MCID has its associated pitfalls, and none has consistently proven to be superior. The applied methodology should be selected according to the characteristics of the data and the disease under study [23]. Additionally, the role of the FACT-Hep for assessing outcomes after surgical resection of HCC has not been robustly studied or validated. As such, the responsiveness of the HRQoL outcomes for this population subset is not clear; for example, the reliability (and change over time) of emotional, psychological, and social responses after surgical resection of HCC needs further study.

## 5. Conclusions

In conclusion, this study revealed that HRQoL scores are independent predictors of MCID and overall survival after surgical resection of HCC. The MCID values were 3.6 for the SF-36 PCS; 4.2 for the SF-36 MCS; 5.4 for the FACT-G total score; and 6.7 for the FACT-Hep total score. The findings of this study may be useful for preoperative management of patient expectations and for developing shared decision-making measures for patients undergoing surgical resection of HCC.

## Figures and Tables

**Figure 1 jcm-08-00576-f001:**
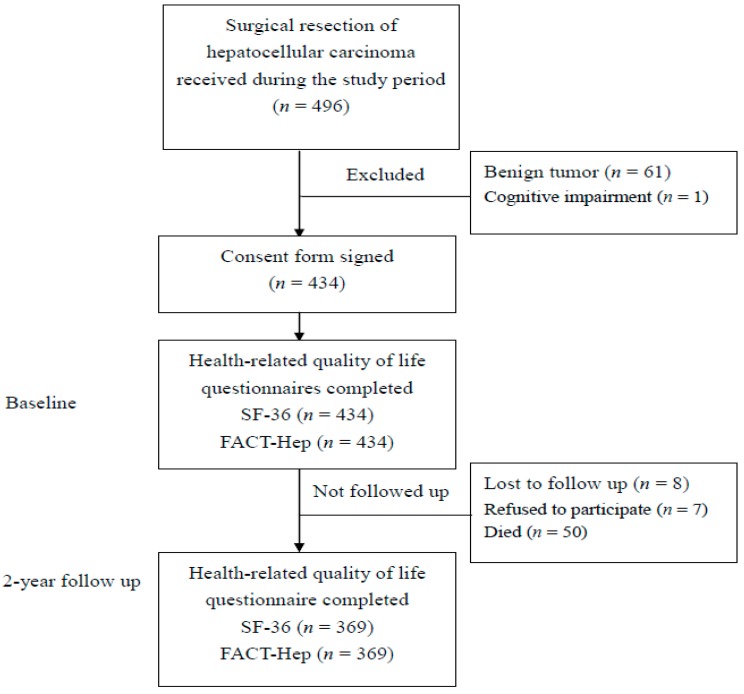
Flow chart showing population changes during the study, including subjects who met initial exclusion criteria, those who later declined to participate and those who lost to follow-up. SF-36: 36-Item Short Form Survey; FACT-Hep: Functional Assessment of Cancer Therapy-Hepatobiliary.

**Table 1 jcm-08-00576-t001:** Demographic and clinical characteristics of 369 patients with hepatic resection for hepatocellular carcinoma.

Variable		N (%) or Mean ± SD
Gender	Male	271 (73.4)
	Female	98 (26.6)
Age, years		60.2 ± 10.8
Marital status	Married	335 (90.8)
	Divorced or widowed	34 (9.2)
Education		8.7 ± 3.6
	No formal education	26 (7.1)
	Primary school	122 (33.1)
	Junior high school	75 (20.3)
	Senior high school	92 (24.9)
	College or above	54 (14.6)
Body mass index, kg/m^2^		25.0 ± 3.5
	Normal (18.5~24.9 kg/m^2^)	218 (59.1)
	Overweight (25.0~29.9 kg/m^2^)	124 (33.6)
	Obese (≥30.0 kg/m^2^)	27 (7.3)
Charlson co-morbidity index, score		1.6 ± 1.3
Co-residence with family	Yes	358 (97.0)
	No	11 (3.0)
Smoking	Yes	71 (19.2)
	No	298 (80.8)
Drinking	Yes	78 (21.1)
	No	291 (78.8)
Tumor stage	I	216 (58.6)
	II	102 (27.6)
	III	51 (13.8)
Chemotherapy	Yes	11 (3.0)
	No	358 (97.0)
Radiotherapy	Yes	5 (1.4)
	No	364 (98.6)
Average length of stay, days		13.0 ± 6.6

SD: standard deviation.

**Table 2 jcm-08-00576-t002:** Mean ± standard deviation for SF-36 and Functional Assessment of Cancer Therapy-Hepatobiliary (FACT-Hep) before and after resection for hepatocellular carcinoma (*n* = 369) *.

Variable	Preoperative	2 Years Postoperative	2 Years Postoperative-Preoperative	*p* Value
SF-36 PCS	56.0 ± 8.7	61.7 ± 9.5	5.8 ± 7.2	*p* < 0.001
SF-36 MCS	48.5 ± 8.1	57.1 ± 9.8	8.5 ± 8.4	*p* < 0.001
FACT-G total	91.2 ± 10.4	98.7 ± 10.8	7.5 ± 10.9	*p* < 0.001
FACT-Hep total	156.9 ± 14.2	165.6 ± 15.8	9.7 ± 13.3	*p* < 0.001

* Both PCS and MCS scores were converted to obtain a mean of 50 and a standard deviation of 10 compared to the normal (nationwide) group. SF-36, 36-Item Short Form Survey; PCS, physical component summary; MCS, mental component summary; FACT-G, Functional Assessment of Cancer Therapy-General; FACT-Hep, Functional Assessment of Cancer Therapy-Hepatobiliary.

**Table 3 jcm-08-00576-t003:** Odds of achieving minimal clinical important difference (MCID) in health-related quality of life according to multivariate logistic regression model *.

Variables	Odds Ratio (95% CI)	*p* Value
SF-36 PCS		
Preoperative SF-36 PCS score	0.90 (0.84, 0.96)	<0.001
Age	1.05 (1.01, 1.10)	0.041
Gender (male vs. female)	0.31 (0.12, 0.83)	0.019
Education	1.14 (1.01, 1.30)	0.040
Body mass index	1.11 (1.09, 1.12)	0.045
SF-36 MCS		
Preoperative SF-36 MCS score	0.80 (0.73, 0.88)	<0.001
Age	1.01 (1.01, 1.02)	<0.001
Body mass index	0.91 (0.84, 0.99)	0.034
Charlson co-morbidity index	1.53 (1.13, 1.94)	<0.001
FACT-G total		
Preoperative FACT-G total score	0.92 (0.90, 0.95)	<0.001
Age	1.04 (1.01, 1.07)	0.007
Education	1.12 (1.04, 1.21)	0.002
Body mass index	1.19 (1.09, 1.29)	<0.001
FACT-Hep total		
Preoperative FACT-Hep total score	0.97 (0.95, 0.98)	<0.001
Age	1.05 (1.02, 1.07)	0.001
Education	1.11 (1.03, 1.19)	0.007
Body mass index	1.09 (1.01, 1.18)	0.020

* The full model was adjusted for preoperative functional status, gender, age, marital status, education, body mass index, Charlson co-morbidity index, co-residence with family, smoking, drinking, tumor stage, chemotherapy, radiotherapy, and average length of stay. SF-36, 36-Item Short Form Survey; PCS, physical component summary; MCS, mental component summary; FACT-G, Functional Assessment of Cancer Therapy-General; FACT-Hep, Functional Assessment of Cancer Therapy-Hepatobiliary.

**Table 4 jcm-08-00576-t004:** Overall survival analysis by Cox multivariable proportional hazard regression model *.

Variable	HR (95% CI)	*p* Value
Preoperative SF-36 PCS score	1.05 (1.03, 1.08)	<0.001
Preoperative SF-36 MCS score	1.03 (1.01, 1.05)	<0.001
Preoperative SF-36 physical function	1.06 (1.01, 1.10)	<0.001
Preoperative SF-36 role physical	1.03 (1.00, 1.05)	<0.001
Preoperative SF-36 bodily pain	1.02 (1.01, 1.03)	0.008
Preoperative SF-36 general health	1.07 (1.01, 1.14)	<0.001
Preoperative SF-36 vitality	1.02 (1.01, 1.04)	0.001
Preoperative SF-36 social function	1.02 (1.01, 1.03)	0.003
Preoperative SF-36 role emotional	1.04 (1.01, 1.06)	<0.001
Preoperative SF-36 mental health	1.03 (1.00, 1.05)	<0.001
Preoperative FACT physical well-being	1.04 (1.00, 1.07)	<0.001
Preoperative FACT social/family well-being	1.01 (1.01, 1.02)	0.010
Preoperative FACT functional well-being	1.03 (1.01, 1.06)	<0.001
Preoperative FACT emotional well-being	1.03 (1.01, 1.05)	<0.001
Preoperative FACT additional concerns	1.02 (1.00, 1.04)	<0.001
Preoperative FACT-G total score	1.07 (1.01, 1.14)	<0.001
Preoperative FACT-Hep total score	1.10 (1.02, 1.19)	<0.001
Education	1.10 (1.02, 1.18)	0.012
Body mass index	1.02 (1.01, 1.04)	0.002
Charlson co-morbidity index	0.83 (0.70, 0.99)	0.040

* The full model was adjusted for preoperative functional status, gender, age, marital status, education, body mass index, Charlson co-morbidity index, co-residence with family, smoking, drinking, tumor stage, chemotherapy, radiotherapy, and average length of stay. SF-36, 36-Item Short Form Survey; PCS, physical component summary; MCS, mental component summary; FACT-G, Functional Assessment of Cancer Therapy-General; FACT-Hep, Functional Assessment of Cancer Therapy-Hepatobiliary; HR, hazard ratio; CI, confidence interval.

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
