# Peer review of "Preoperative Health-Related Quality of Life Predicts Minimal Clinically Important Difference and Survival after Surgical Resection of Hepatocellular Carcinoma"

_jcm, 2019, doi:10.3390/jcm8050576_

Round 1
Reviewer 1 Report
The study tried to defined MCID and evaluated the use of preoperative HRQoL for predicting MCID and survival after surgical resection of hepatocellular carcinoma (HCC). The study was performed in three tertiary academic hospitals and based on a 2 years follow-up.
The study could be of interest for the people involved in the HCC managmnet but there are some issue that should be discussed:
- The authors reported in the introduction that “A clinically important difference is a change that a patient or clinician would consider meaningful or worthwhile……..Measures of clinical significance such as the minimal clinically important difference (MCID) are increasingly used as a standard clinical outcome measure. The MCID is defined as the smallest outcome change that the patient perceives as clinically important.”
However the authors found that patients with a high preoperative score did not achieve a postoperative MCID in the outcome measure. They explain this lack of “efficacy” reporting that patients who already have high HRQoL scores before surgery has less potential for achieving a HRQoL score improvement that meets the criteria for an MCID.
The fact that in a group of patients the MCID is not applicable, raise some questions to its utility in assessing the efficacy of Hepatic resection.
The authors should better explain the meaning of assessing the MCID and how to mange the different applicability in the patients with high preoperative score compare to those with low preoperative score.
- The authors showed in the artcle how high preoperative HRQoL score had significant positive associations overall survival, however they not assessed if one of the different HRQoL score used in the study is better compare to the other.
Having information on which HRQoL parameters is beeter to use to predict the patients survival could be more useful.
The authors should include this type of analysis in the study.
- The authors should explain how they manage the missing quality of life data at 2 yers for the patients lost or dead during follow up.
Author Response
Comments and Suggestions for Authors
The study tried to defined MCID and evaluated the use of preoperative HRQoL for predicting MCID and survival after surgical resection of hepatocellular carcinoma (HCC). The study was performed inthree tertiary academic hospitals and based on a 2 years follow-up.
The study could be of interest for the people involved in the HCC managmnet but there are some issue that should be discussed:
- The authors reported in the introduction that “A clinically important difference is a change that a patient or clinician would consider meaningful or worthwhile……..Measures of clinical significance such as the minimal clinically important difference (MCID) are increasingly used as a standard clinical outcome measure. The MCID is defined as the smallest outcome change that the patient perceives as clinically important.”
However the authors found that patients with a high preoperative score did not achieve a postoperative MCID in the outcome measure. They explain this lack of “efficacy” reporting that patients who already have high HRQoL scores before surgery has less potential for achieving a HRQoL score improvement that meets the criteria for an MCID.
The fact that in a group of patients the MCID is not applicable, raise some questions to its utility in assessing the efficacy of Hepatic resection.
The authors should better explain the meaning of assessing the MCID and how to mange the different applicability in the patients with high preoperative score compare to those with low preoperative score.
Ans:
Thank you for your comment. The aim of this study was to calculate and report MCID values for commonly used HRQoL scales. Changes in HRQoL over time and/or by treatment may not correlate with the direction of change (positive vs. negative) or with the magnitude of change in clinical improvements in outcomes as perceived by patient. Furthermore, cancer treatment has a greater impact on the HRQoL of HCC surgery patients compared to changes in MCID values (SF-36 PCS, SF-36 MCS and FACT-G total score) differed across HRQoL domains, and they differed according to the improvement and deterioration perceived by the patient. For the patients in this study, scores related to physical function and emotional function domains were lower before treatment than at the time of treatment and immediately after treatment. Additionally, the anxiety and depression caused by the diagnosis and initiation of treatment, which they reported in the initial questionnaire, may have been subsequently ameliorated by their increased familiarity with the treatment and by the supportive psychosocial care provided by clinical service teams. Reduced anxiety and depression might explain why patients with a low preoperative score had a postoperative MCID that was classified as higher “efficacy” in comparison with the postoperative MCID obtained by patients with a high preoperative score, who already had high familiarity with the treatment and had strong psychosocial support. The above statements have been added to the Discussion section of the revised manuscript (lines 8-19, page 13 & lines 1-6, page 14). Again, thank you.
- The authors showed in the artcle how high preoperative HRQoL score had significant positive associations overall survival, however they not assessed if one of the different HRQoL score used in the study is better compare to the other.
Having information on which HRQoL parameters is beeter to use to predict the patients survival could be more useful.
The authors should include this type of analysis in the study.
Ans:
Following for your comment above, we included different HRQoL parameters to analyze overall survival analysis in Table 4. The statements also are added in the Results section (lines 4-5, page 12) and in the Discussion section (lines 17-19, page 12 & lines 1-2, page 13) to clarify the issue. Thank you.
- The authors should explain how they manage the missing quality of life data at 2 yers for the patients lost or dead during follow up.
Ans:
To address the concerns of the reviewer, the following text has been added to the Results section of the revised manuscript (lines 2-7, page 10).
Subjects who remained in the study throughout the 2-year period and those who died or were lost to follow up between baseline and the 2nd year after discharge did not statistically differ in gender, age, marital status, education, BMI, CCI score, co-residence with family, smoking, drinking, tumor stage, chemotherapy, radiotherapy, ALOS, or in any of the aforementioned preoperative HRQoL parameters (data not shown).

Reviewer 2 Report
Although treatment regimens, from the anti-cancer agents to the state-of-art surgical technique, are clearly improved, the prediction of clinical outcome or prognosis by such treatments did not improve. Nevertheless, there are many cases where the management of various ineffective events associated with treatment is not accompanied by efforts to improve the quality of life of cancer patients. This study provides critical awareness to physicians on this issue. The authors also carefully considered the attitudes of cancer patients receiving surgical resection. One side, it might be to judge as a simple article describing the results of the questionnaire, on the other side, it was important meaning to the management of quality of life is analyzed in point of view of cancer patients. This reviewer recommend that it is more important from the latter point between the above views. Even so, one concern is raised. Although there are several ways or alternatives to improve the quality of life, there is no paragraph introducing the development efforts for recent management of quality of life.
Author Response
Comments and Suggestions for Authors
Although treatment regimens, from the anti-cancer agents to the state-of-art surgical technique, are clearly improved, the prediction of clinical outcome or prognosis by such treatments did not improve. Nevertheless, there are many cases where the management of various ineffective events associated with treatment is not accompanied by efforts to improve the quality of life of cancer patients. This study provides critical awareness to physicians on this issue. The authors also carefully considered the attitudes of cancer patients receiving surgical resection. One side, it might be to judge as a simple article describing the results of the questionnaire, on the other side, it was important meaning to the management of quality of life is analyzed in point of view of cancer patients. This reviewer recommend that it is more important from the latter point between the above views. Even so, one concern is raised. Although there are several ways or alternatives to improve the quality of life, there is no paragraph introducing the development efforts for recent management of quality of life.
Ans:
Thank you for your comments. As advised, the following text has been added to address recent developments in patient quality of life management.
Quality of life is widely used in the international development literature because it provides analysis of development on a measure broader than the standard of living. According to development theory, however, what constitutes desirable change for a particular society is debatable, a desirable change for one society may not be considered desirable for another society, and the different ways that quality of life is defined by institutions therefore shapes how these organizations work for its improvement as a whole. In cancer research, the Functional Assessment of Cancer Therapy-Hepatobiliary (FACT-Hep) measure is one of the most widely used patient-reported questionnaires for measuring HRQoL. Unfortunately, studies that have used the FACT-Hep rarely report clinically significant differences, even though guidelines for assessing clinical significant differences have already been established.
We added these statements in the Introduction in the revised manuscript (lines 9-19, page 3). Again, thank you.

Round 2
Reviewer 1 Report
The changes made by the authors have improved the paper substantially.
I have no other comments.